# MotionRAG: Motion Retrieval-Augmented Image-to-Video Generation

**Chenhui Zhu**[1]  **Yilu Wu**[1]  **Shuai Wang**[1]  **Gangshan Wu**[1]  **Limin Wang**[1,2]*

[1]State Key Laboratory for Novel Software Technology, Nanjing University
[2]Shanghai AI Laboratory
`https://github.com/MCG-NJU/MotionRAG`

## Abstract

Image-to-video generation has made remarkable progress with the advancements in diffusion models, yet generating videos with realistic motion remains highly challenging. This difficulty arises from the complexity of accurately modeling motion, which involves capturing physical constraints, object interactions, and domain-specific dynamics that are not easily generalized across diverse scenarios. To address this, we propose MotionRAG, a retrieval-augmented framework that enhances motion realism by adapting motion priors from relevant reference videos through Context-Aware Motion Adaptation (CAMA). The key technical innovations include: (i) a retrieval-based pipeline extracting high-level motion features using video encoder and specialized resamplers to distill semantic motion representations; (ii) an in-context learning approach for motion adaptation implemented through a causal transformer architecture; (iii) an attention-based motion injection adapter that seamlessly integrates transferred motion features into pretrained video diffusion models. Extensive experiments demonstrate that our method achieves significant improvements across multiple domains and various base models, all with negligible computational overhead during inference. Furthermore, our modular design enables zero-shot generalization to new domains by simply updating the retrieval database without retraining any components. This research enhances the core capability of video generation systems by enabling the effective retrieval and transfer of motion priors, facilitating the synthesis of realistic motion dynamics.

## 1 Introduction

Recent advancements in generative models have revolutionized image-to-video synthesis, enabling the creation of short video clips from static images with unprecedented visual quality [1, 2, 3, 4, 5]. These models, primarily based on diffusion architectures [6], excel at preserving the appearance of input images while introducing temporal dynamics. However, despite their impressive visual fidelity, a critical challenge persists: generating physically plausible and semantically coherent motion remains a significant and unresolved issue [7, 8].

The core challenge stems from the inherent complexity of motion. Unlike appearance, which can be directly inferred from a single frame, motion involves capturing physical constraints, object interactions, and domain-specific dynamics, making its modeling significantly more difficult. Existing methods typically rely on end-to-end training on large video datasets [5, 3, 9], where motion knowledge develops naturally through exposure to various examples. While this approach yields improvements, it struggles with compositional generalization—the ability to combine familiar elements in novel ways, such as "*an astronaut riding a horse on the moon.*"

---

*Corresponding author: `lmwang@nju.edu.cn`

39th Conference on Neural Information Processing Systems (NeurIPS 2025).

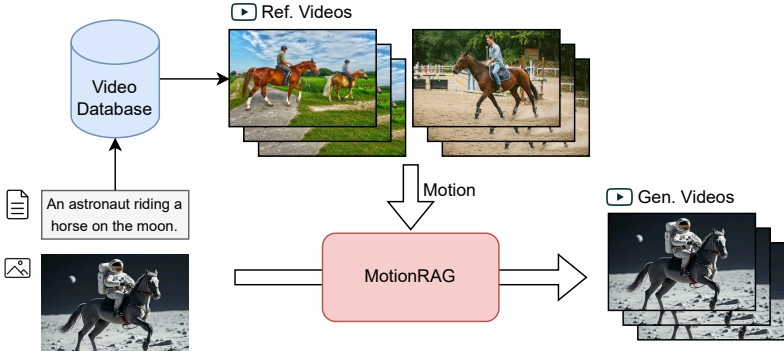

Figure 1: **Illustration of cross-domain motion transfer.** Our approach retrieves videos of people riding horses and transfers their motion priors to generate an astronaut riding a horse on the moon, while preserving the appearance characteristics of the input image.

A key insight driving our research is that **motion can be inherently transferred across domains** [10, 11]. For example, the motion of a person riding a horse can be applied to generate an astronaut riding a horse, despite significant differences in visual appearance, as shown in Figure 1. This transferability is due to the physical and kinematic constraints governing motion, which remain consistent even when surface appearances change [12]. However, effectively extracting and transferring this motion is challenging, as motion features are often mixed with appearance information in video representations [13]. Current image-to-video models primarily rely on text descriptions to infer motion dynamics, but textual descriptions inherently lack the precise temporal coordination and kinematic details that actual video examples provide. This fundamental limitation motivates our retrieval-augmented approach, which leverages real video motion patterns to guide generation with richer and more physically plausible dynamics.

To tackle these challenges, we introduce a novel retrieval-augmented framework **MotionRAG** for image-to-video generation that enhances motion realism through explicit cross-domain transfer. Our approach comprises three key components: (i) a text-based retrieval system that identifies videos with relevant motion, (ii) a context-aware motion adaptation (CAMA) module that adapts the extracted motion information to the target image, and (iii) a motion-guided generation process that synthesizes the final video while preserving appearance fidelity. The core technical innovation of our work lies in our Context-Aware Motion Adaptation (CAMA) module, which formulates motion transfer as an in-context learning problem [14, 15]. Drawing inspiration from recent advances in LLM, our transformer-based architecture processes a sequence of retrieved examples to infer appropriate motion priors for the target image. By arranging examples in reverse similarity order and employing causal attention, the model effectively learns to adapt motion features across visual domains without requiring domain-specific fine-tuning.

Through our experiments, we demonstrate that our approach significantly improves motion realism across multiple domains and base models. Our method achieves consistent quality improvements when integrated with various state-of-the-art image-to-video models. Quantitative and qualitative evaluations confirm that videos generated with our method exhibit more natural and physically plausible motion compared to existing approaches.

Our contributions can be summarized as follows: (i) We introduce MotionRAG, a retrieval-augmented framework that extracts and transfers high-level motion representations from semantically relevant videos to guide image animation. (ii) We propose CAMA, a novel in-context learning approach for motion transfer that adapts motion patterns across visual domains without requiring per-instance fine-tuning. (iii) Extensive experiments demonstrate our method significantly improves motion quality across multiple baseline models with negligible computational overhead, enhancing even state-of-the-art models by substantial margins.

## 2 Related Work

**Retrieval-Augmented Generation.** Retrieval-Augmented Generation (RAG)[16] is a powerful approach that improves pretrained models by retrieving relevant information from external sources

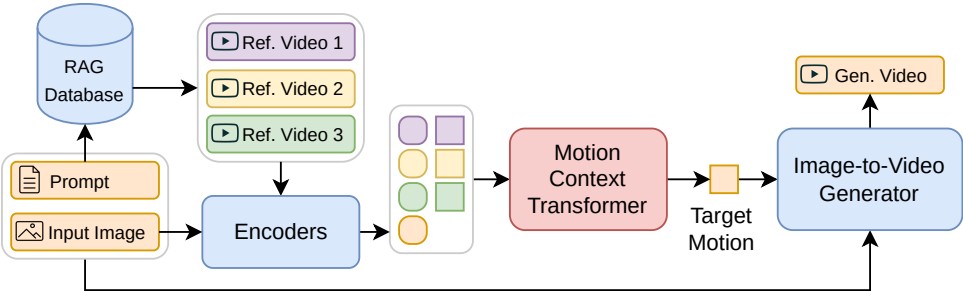

Figure 2: **Our MotionRAG framework.** Text prompts retrieve relevant videos from a database. Motion information from these references are adapted to the input image via our Motion Context Transformer, then injected into an image-to-video generator to produce the final output.

during generation. Originally designed for natural language processing, RAG allows models to access domain-specific knowledge on demand, leading to more accurate and relevant outputs[17]. Inspired by its success in language tasks, similar methods have been applied to visual domains. For example, ImageRAG [18] retrieves related images to improve image generation quality. In video generation, methods like search-T2V [19] follow a similar idea by using a video database as a motion prior. This turns text-to-video (T2V) generation into a search-based process, boosting performance without requiring large-scale training. However, search-T2V only supports text-to-video generation and cannot use a reference image as input, limiting its use in image-to-video tasks. It also requires expensive fine-tuning for each generation, which takes several minutes. In contrast, our method uses in-context learning to adapt motion patterns quickly and efficiently without extra training.

**Video Motion Customization.** Video customization aims to adapt pre-trained video generation models to personalized concepts, often by fine-tuning on reference videos [20, 21, 22, 23]. For example, MotionDirector [20] used a dual-path design with a temporal objective, VMC [21] employed inter-frame residuals to distill motion, and DreamVideo [22] applied adapters to separate motion and appearance. Customize-A-Video [23] enhanced appearance features and used temporal LoRA [24] for motion learning. Unlike these methods, which require model fine-tuning for each video, our retrieval-based approach uses in-context learning to adapt motion without changing model parameters. This enables efficient generalization across domains and allows combining motion priors from multiple retrieved examples for more flexible and context-aware generation.

**Image Animation.** Animating a single image has received increasing attention. Several diffusion-based methods [2, 25, 26, 27] have been proposed for open-domain image animation. Stable Video Diffusion [1] introduced a latent diffusion model with multi-stage training for high-resolution video generation. DynamiCrafter [2] projected images into a text-aligned space using a query transformer to better preserve visual details. I2VGen-XL [25] improved clarity and continuity via a two-stage design that decouples semantic and temporal modeling. VideoComposer [26] enabled controllable generation by encoding spatial-temporal conditions. MoG [28] used explicit motion guidance for high-fidelity frame interpolation. Motion-I2V [27] relied on optical flow to improve motion consistency. Unlike Motion-I2V which relies on low-level optical flow, our method extracts and injects high-level semantic motion features from retrieved references. These high-level motion representations capture more abstract dynamics that are easier to transfer across different visual domains and subject appearances. Additionally, our approach extracts these high-level motion representations in less than one second, compared to the several minutes required for optical flow generation in Motion-I2V, making our method substantially more efficient for practical applications.

## 3 Methodology

### 3.1 Framework Overview

To tackle the motion realism challenge in image-to-video generation, we propose a novel Motion Retrieval Augmented image-to-video Generation (**MotionRAG** for brevity) framework. Our approach uses a simple yet effective three-stage process(retrieval, adaptation, synthesis) to improve motion quality in generated videos, as shown in Figure 2. Given an input image $\mathbf{I} \in \mathbb{R}^{h \times w \times 3}$, and a text prompt $\mathbf{T} \in \mathbb{R}^{N \times d_t}$ with $N$ tokens and embedding dimension $d_t$, our framework operates as follows:

First, we perform text-based retrieval to identify the most contextually relevant video samples $\{\mathbf{V}_i\}_{i=1}^K$ from a comprehensively indexed database. Subsequently, our Context-Aware Motion Adaptation component transforms these retrieved motion patterns into target image compatible features. Denoting the motion feature extraction function as $f_m$ and the image encoder as $f_i$, we compute the adapted motion representation $\hat{\mathbf{M}}$ as:

$$\hat{\mathbf{M}} = \mathcal{T}\big(f_m(\{\mathbf{V}_i\}_{i=1}^K), f_i(\{\mathbf{F}_i\}_{i=1}^K, \mathbf{I})\big), \tag{1}$$

where $\mathbf{F}_i$ represents the first frame of retrieved video $\mathbf{V}_i$, and $\mathcal{T}$ denotes our motion context transformer that adapts retrieved motion patterns to align with the visual characteristics of the target image. In the synthesis stage, we generate the output video $\hat{\mathbf{V}} \in \mathbb{R}^{T \times h \times w \times 3}$ with $T$ frames by conditioning a diffusion-based generator $\mathcal{G}$ on the input image, text prompt, and adapted motion features:

$$\hat{\mathbf{V}} = \mathcal{G}(\mathbf{I}, \mathbf{T}, \hat{\mathbf{M}}). \tag{2}$$

This principled approach enables the generation of videos with enhanced motion fidelity while maintaining visual coherence with the input image and semantic alignment with the text description. By explicitly incorporating real-world motion priors through retrieval-augmented generation, our framework improves temporal dynamics and physical plausibility in the synthesized videos.

## 3.2 Text-based Video Retrieval

Text-based video retrieval provides relevant motion exemplars with high quality as references. Our retrieval pipeline comprises two interconnected components: database construction and semantic retrieval. To achieve robust and accurate retrieval, we chose to construct a retrieval database with text embedding as the retrieval index.

**Retrieval Database Construction.** We curate a diverse video dataset with associated captions. To keep simplicity, we encode the corresponding captions with embedding techniques (e.g., Sentence-BERT [29]) to generate a dense representation $\mathbf{e}_j \in \mathbb{R}^d$ as the retrieval index.

**Semantic Retrieval.** These embeddings are indexed using approximate nearest neighbor techniques to facilitate efficient retrieval during inference. Input text prompt $\mathbf{T}$ undergoes identical encoding to produce query embedding $\mathbf{e}_q$. We then compute the semantic similarity between this query and all database entries using cosine similarity:

$$sim(\mathbf{V}_j, \mathbf{T}) = \frac{\mathbf{e}_j \cdot \mathbf{e}_q}{\|\mathbf{e}_j\| \|\mathbf{e}_q\|}. \tag{3}$$

The system subsequently retrieves the top-$K$ videos $\{\mathbf{V}_i\}_{i=0}^K$ ordered by descending similarity scores, where $i = 0$ represents the most semantically aligned exemplar. This retrieval mechanism provides a foundation for our motion adaptation process by identifying real-world motion patterns that exhibit strong semantic alignment with the desired video content, thereby establishing a crucial bridge between text prompts and motion representations.

## 3.3 Context-Aware Motion Adaptation

To effectively transfer motion priors from retrieved videos to the target image, we propose Context-Aware Motion Adaptation. Figure 3 illustrates our approach, which operates through three modules.

**Motion Feature Extraction.** We leverage the pretrained VideoMAE [30] encoder to extract high-level motion representations from each retrieved video $\mathbf{V}_k$. Unlike conventional optical flow that captures low-level pixel trajectories, these features encapsulate semantic motion patterns. The VideoMAE encoder processes each video $\mathbf{V}_k$ to produce dense spatio-temporal features, which are then processed through a learnable resampler [31] module that distills these representations into a compact set of $L$ motion tokens: $f_m(\mathbf{V}_k) \in \mathbb{R}^{L \times d}$. This approach enables us to capture coherent motion patterns while discarding appearance-specific details that might hinder effective transfer across visual domains.

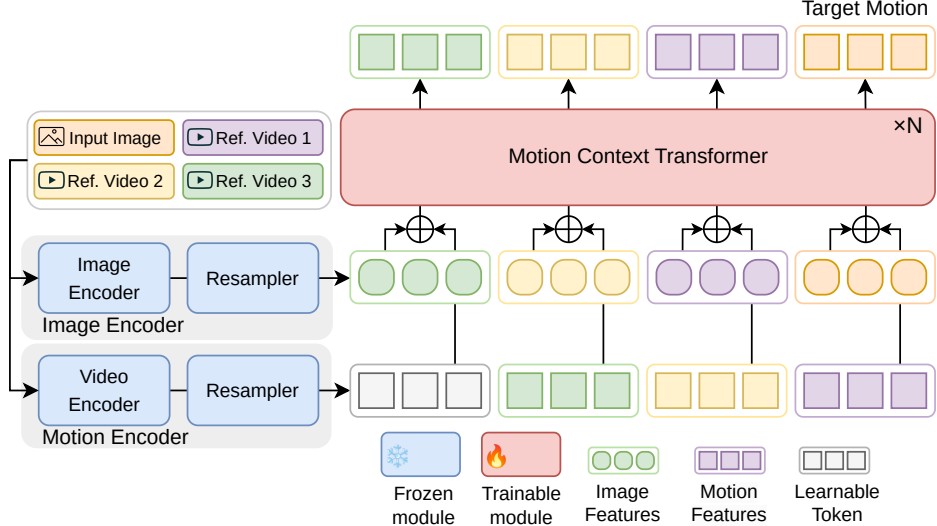

Figure 3: **Context-Aware Motion Adaptation (CAMA) architecture.** Appearance and motion features from retrieved videos and the target image are processed through a causal transformer, which learns to predict appropriate motion features for the target image through in-context learning.

**Image Feature Extraction.** For appearance encoding, we utilize the DINO [32] vision transformer to process both the target image $\mathbf{I}$ and the first frame $\mathbf{F}_k$ of each retrieved video. These visual inputs are encoded and subsequently compressed through a parallel resampler architecture to obtain appearance features $f_i(\mathbf{F}_k) \in \mathbb{R}^{L \times d}$ and $f_i(\mathbf{I}) \in \mathbb{R}^{L \times d}$. We specifically design both appearance and motion resamplers to output the same token count $L$ and feature dimension $d$, facilitating direct feature addition when constructing the transformer input sequence.

**Adaptive Motion Transfer.** To transfer motion features to the target image while preserving motion semantics, we introduce a causal transformer architecture that performs in-context learning of motion-appearance relationships. The retrieval system returns videos ordered by descending relevance $\{\mathbf{V}_1, \mathbf{V}_2, \ldots, \mathbf{V}_K\}$, where $\mathbf{V}_1$ represents the most semantically relevant video. We arrange these examples in reverse retrieval order: $\{\mathbf{F}_K, \mathbf{F}_{K-1}, \ldots, \mathbf{F}_1, \mathbf{F}_0\}$, where $\mathbf{F}_0$ represents the target image. This reverse ordering serves a crucial purpose: it allows the model to first process less relevant examples, gradually transitioning to more relevant ones, and finally to the target image. This progressive ordering enables the transformer to build a more refined understanding of motion-appearance relationships before addressing the target image. For each retrieved video frame $\mathbf{F}_n$ and its corresponding motion feature $f_m(\mathbf{V}_n)$, we construct input tokens $\mathbf{X}_n = f_i(\mathbf{F}_n) + f_m(\mathbf{V}_{n+1})$ by directly adding feature representations. The Motion Context Transformer (MCT) processes these examples sequentially, with a critical constraint on the attention mechanism: tokens corresponding to each video frame can only attend to tokens from the same video and tokens from previously processed videos in the sequence. This causal attention mask ensures that predictions rely only on previously observed examples, which is essential for maintaining the in-context learning paradigm, where the model learns patterns from example pairs without "peeking" at future examples.

By positioning the target image $\mathbf{F}_0$ last in the sequence, the model accumulates sufficient context from all retrieved examples before generating motion features for the target. The transformer infers compatible motion features $\hat{\mathbf{M}}$ by extrapolating from the learned examples, effectively adapting motion patterns from retrieved videos to the visual characteristics of the target image. This approach enables our model to rapidly adapt to new visual domains without requiring explicit fine-tuning, as it learns to transfer motion patterns across varying appearance contexts through in-context learning.

### 3.4 Motion-Guided Video Generation

At the core of our approach is a conditional diffusion model for video generation. Diffusion models [6, 33] operate by gradually denoising a random Gaussian noise through a series of denoising steps. The

forward process progressively adds noise to the data, while the reverse process learns to denoise and recover the original data distribution. In conditional image-to-video diffusion models, this reverse process is typically guided by an image $\mathbf{I}$ and text prompt $\mathbf{T}$, formulated as:

$$p_\theta(\mathbf{x}_0|\mathbf{x}_t, \mathbf{I}, \mathbf{T}) = \mathcal{N}(\mu_\theta(\mathbf{x}_t, t, \mathbf{I}, \mathbf{T}), \Sigma_\theta(\mathbf{x}_t, t)), \tag{4}$$

where $\mathbf{x}_t$ represents the noisy latent at diffusion timestep $t$, and $\mu_\theta$ is the predicted noise mean parameterized by UNet [1, 2, 34] or DiT [5, 35, 36]. Our approach extends this framework by incorporating the transferred motion features $\hat{\mathbf{M}}$ as an additional conditioning signal:

$$p_\theta(\mathbf{x}_0|\mathbf{x}_t, \mathbf{I}, \mathbf{T}, \hat{\mathbf{M}}) = \mathcal{N}(\mu_\theta(\mathbf{x}_t, t, \mathbf{I}, \mathbf{T}, \hat{\mathbf{M}}), \Sigma_\theta(\mathbf{x}_t, t)), \tag{5}$$

To incorporate the transferred motion features into the diffusion process, we employ an adapter-based injection mechanism inspired by IP-Adapter [37]. Our approach, which we call Motion-Adapter, integrates motion information without requiring modifications to the underlying generative model architecture. The motion-guided video generation process starts with a pretrained image-to-video diffusion model. We strategically insert our Motion-Adapter modules after each cross-attention in the UNet or DiT backbone, allowing motion features to guide generation after text conditioning has been applied. These adapters modify the latent representations to follow the desired motion pattern while maintaining appearance consistency with the input image.

Formally, given the hidden features $\mathbf{Z}_i$ at the $i$-th layer, which have already incorporated text conditioning through the text cross-attention mechanism, we compute:

$$\mathbf{Z}'_i = \mathbf{Z}_i + \text{Attention}(\mathbf{Q}_i, \mathbf{K}_i, \mathbf{V}_i), \tag{6}$$

where: (i) $\mathbf{Q}_i = \mathbf{Z}_i\mathbf{W}_i^q$ represents queries derived from the text-conditioned visual tokens. (ii) $\mathbf{K}_i = \hat{\mathbf{M}}\mathbf{W}_i^k$ provides keys from our motion features. (iii) $\mathbf{V}_i = \hat{\mathbf{M}}\mathbf{W}_i^v$ supplies values from the same motion features. The projection matrices $\mathbf{W}_i^q$, $\mathbf{W}_i^k$, and $\mathbf{W}_i^v$ are learnable parameters specific to each layer. Our approach preserves all original pretrained weights of the diffusion model, which remain frozen during training. Only the adapter-specific parameters are optimized, allowing our method to leverage the full generative capabilities of the base model while introducing motion control without catastrophic forgetting of pretrained knowledge.

### 3.5 Training and Inference

Our training strategy employs a two-stage approach to effectively learn motion patterns and their domain-adaptive transfer. In the first stage, we train the Motion-Adapter and motion resampler modules using ground truth videos. Given a video $\mathbf{V}$ and its first frame $\mathbf{F}^0$, we extract motion features using video encoder and train the motion resampler to produce compact, semantically-rich representations. Concurrently, the Motion-Adapter is trained to condition the diffusion model using these representations to reconstruct the original video. This stage ensures our adapter can effectively inject motion information into the generation process. In the second stage, we freeze both the motion resampler while training the Motion Context Transformer and image resampler. For each training video, we extract motion tokens $f_m(\mathbf{V})$ using the frozen resampler and construct in-context learning sequences by sampling similar videos from our database. The transformer is trained to predict the motion features of the target video using an L2 loss between predicted features $\hat{\mathbf{M}}$ and ground truth features $f_m(\mathbf{V})$:

$$\mathcal{L}_{transfer} = \|\hat{\mathbf{M}} - f_m(\mathbf{V})\|_2^2. \tag{7}$$

During inference, our framework operates end-to-end without requiring domain-specific fine-tuning. Given a text prompt and image, we retrieve semantically similar videos from our database and process them through the Motion Context Transformer to predict adapted motion representations. These features are then injected into the generation model via the Motion-Adapter to produce the final video that exhibits the desired motion patterns while preserving the appearance of the reference image. A key advantage of our approach is its extensibility to new domains without parameter updates. To generate videos in specialized domains (e.g., instructional videos, scientific visualizations), one only needs to construct a new retrieval database containing examples from the target domain. The model's in-context learning capabilities enable it to adapt motion patterns across substantial domain shifts without explicit fine-tuning of any parameters, significantly enhancing its practical utility.

# 4 Experiments

## 4.1 Implementation Details

**Video Generation Models.** We implement our method on three image-to-video generation models: Stable Video Diffusion (SVD) [1], Dynamicrafter [2], and CogVideoX-5b [5]. For SVD and Dynamicrafter, we generate 16-frame videos at 1024×576 resolution, while CogVideoX-5b produces 17-frame videos at 720×480 resolution. We insert our Motion Adapter modules after every text cross-attention layer in SVD and Dynamicrafter, and after each MMDiT layer in CogVideoX-5b. All adapter modules are trained using AdamW optimizer for approximately 50,000 steps with a batch size of 16 on the OpenVid-1M dataset [38].

**Context-Aware Motion Adaptation.** Our CAMA module utilizes a VideoMAE base model [30] as the video encoder and DINOv2-large [32] as the image encoder. Both Resampler modules employ 4-layer Transformer architectures that distill features into 25 tokens with dimension 1024. The Motion Context Transformer consists of a 4-layer causal Transformer with hidden dimension 1024, trained with AdamW optimizer for 50,000 steps with a batch size of 64. During training, we retrieve the 9 most similar videos from OpenVid-1M to construct the in-context learning examples. Additional implementation details are provided in the supplementary material.

**Retrieval Database.** We construct three video retrieval databases using the GTE-v1.5 model [39] to encode video captions into embedding vectors. Retrieval is performed efficiently via cosine similarity between the query text and the pre-computed caption embeddings. Leveraging a high-performance vector database, this retrieval step is extremely fast, taking only approximately 40ms to query 1-million-entry database on a standard CPU, ensuring it is not a bottleneck in the generation pipeline. Our databases include: (1) **OpenVid-1M** [38]: A large-scale, general-domain video dataset. To create more motion-focused captions suitable for retrieval, we performed a one-time, automated preprocessing step on the original detailed descriptions. We utilized the Llama3.1-8B model to generate concise, motion-centric summaries for each video, a process that is both scalable and easily reproducible. This refined dataset will be made publicly available to facilitate future research. (2) **SkillVid** [40]: A specialized dataset containing instructional and skill-based videos. This database is used to demonstrate our framework's ability to adapt to domain-specific motions. (3) **InternVid-10M** [41]: A massive-scale video-text dataset originally curated for video understanding tasks. Its data distribution (e.g., content, style, captioning focus) differs significantly from our generation-focused datasets. We use it as a challenging out-of-distribution (OOD) database to test the generalization capabilities of our framework.

## 4.2 Datasets and Metrics

We evaluate our method on two datasets: (1) OpenVid-1K, a diverse test set of 1,000 videos sampled from OpenVid-1M [38] with no overlap with the training data, representing general video domains; and (2) SkillVid [40] test set, the test set of SkillVid that we use to assess zero-shot capabilities.

While existing benchmarks like VBench [42] focus primarily on detecting visual artifacts such as flickering, they lack ground truth videos and cannot effectively measure whether generated content exhibits realistic motion patterns. More recent work on physical realism evaluation [43] utilizes low-level metrics like spatial IoU, which are highly effective for simple, controlled scenarios but are less robust for the complex, open-domain videos in our evaluation, where minor, physically plausible variations in motion could be unfairly penalized. Therefore, following prior work [42, 40], we adopt a suite of complementary metrics that holistically assess quality by comparing to ground-truth videos. These include: **Action Score** (semantic motion correctness), **CLIP Score** (frame-level semantic alignment), **DINO Score** (frame-level identity preservation), **FID** [44] (frame visual quality), and **FVD** [45] (overall video quality). Among these, the Action Score is particularly crucial for our task as it evaluates motion from a high-level, semantic perspective, which is more indicative of physical plausibility in complex scenes than pixel-level metrics.

## 4.3 Results on General Domain

Table 1 presents quantitative results on the OpenVid-1K test set. Our retrieval-augmented approach significantly enhances performance across all baseline models. Notably, when applied to CogVideoX,

Table 1: **Quantitative comparison on the OpenVid-1K test set.** Higher is better for Action, CLIP, and DINO scores; lower is better for FVD, FID, and Time (minutes).

| Model | Action↑ | CLIP↑ | DINO↑ | FVD↓ | FID↓ | Time↓ |
|---|---|---|---|---|---|---|
| Cog [5] | 59.9 | 91.2 | 87.8 | 87.1 | 11.8 | 0.99 |
| Cog+RAG | **65.8** | **92.1** | **89.4** | **80.2** | **11.4** | 1.05 |
| DC [2] | 53.5 | 91.0 | 85.8 | 88.4 | 10.9 | 1.46 |
| DC+RAG | **62.1** | **92.3** | **88.4** | **69.0** | **9.7** | 1.49 |
| SVD [1] | 57.5 | 87.2 | 83.6 | **98.0** | 15.7 | 0.74 |
| SVD+RAG | **60.0** | **91.8** | **89.6** | 167.1 | **13.4** | 0.75 |

Table 2: **Zero-shot transfer evaluation on the SkillVid.** Our method demonstrates generalization to instructional videos without any fine-tuning.

| Model | Action↑ | CLIP↑ | DINO↑ | FVD↓ | FID↓ |
|---|---|---|---|---|---|
| Cog[5] | 51.5 | 87.7 | 78.7 | 91.8 | 10.0 |
| Cog+RAG | **53.5** | **89.1** | **81.8** | **83.2** | **9.5** |
| DC [2] | 49.6 | **90.0** | **82.9** | 108.5 | **8.1** |
| DC+RAG | **50.1** | 88.9 | 80.5 | **89.4** | 8.2 |
| SVD[1] | 48.0 | 86.1 | 79.6 | **127.1** | 12.6 |
| SVD+RAG | **49.1** | **90.2** | **85.2** | 130.5 | **10.1** |

our method yields the highest overall Action similarity score of 65.8, representing a substantial 5.9-point improvement that demonstrates RAG's ability to enhance state-of-the-art DiT models.

For Dynamicrafter, our approach increases the Action score by 8.6 points, while SVD shows a 2.5-point improvement. Both CLIP and DINO scores improve consistently across all models, indicating better semantic alignment and identity preservation. The FVD score improvements are particularly striking for CogVideoX (8.0% reduction) and Dynamicrafter (22.0% reduction).

Crucially, these substantial improvements come with negligible computational overhead. Our RAG approach adds only 0.01-0.06 minutes (less than 4 seconds) to inference time across all models, demonstrating the practical viability of our approach for real-world applications. The consistently positive results across diverse model architectures, coupled with the negligible computational cost, underscore that our RAG framework is a highly effective and practical plug-and-play module for enhancing state-of-the-art video generation models.

## 4.4 Zero-Shot Transfer to Specialized Domains

Table 2 demonstrates our method's zero-shot generalization to the SkillVid dataset. Without any fine-tuning, RAG improves motion realism across all models, with CogVideoX showing the most substantial gain (+2.0 points in Action score). Overall video quality improves significantly, with FVD reductions of 9.4% for CogVideoX and 17.6% for Dynamicrafter. SVD benefits particularly in semantic alignment (CLIP +3.8, DINO +5.4), while Dynamicrafter shows modest trade-offs between motion quality and semantic preservation. These results show that our approach effectively transfers to specialized domains by simply changing the retrieval database, requiring no parameter updates.

## 4.5 Ablation Studies

We conduct a detailed analysis of our framework's components using the Dynamicrafter model on the OpenVid-1K dataset. The ablation studies, presented in Tables 3 and 4, systematically validate our design choices. Specifically, we demonstrate that (1) our Motion Context Transformer (MCT) significantly outperforms simpler feature integration strategies, (2) the framework is remarkably robust to noisy context, and (3) it generalizes effectively to an out-of-distribution retrieval database.

**Impact of Motion Adaptation Method.** As shown in Table 3, our Context-Aware Motion Adaptation (CAMA) module is the key to superior performance. While naive approaches like using the top-1 retrieved video (`Top-1`) or averaging features from 9 videos (`Avg-9`) provide moderate gains over the baseline, our Motion Context Transformer (`MCT-9`) dramatically outperforms them. It achieves an Action score of 62.1 (+8.6 over baseline and +4.2 over `Avg-9`) and reduces FVD to 69.0 (-19.4 from baseline and -9.7 from `Avg-9`). This highlights that our method's strength lies not just in retrieval but in the intelligent, in-context synthesis of motion. Remarkably, our performance is very close to the theoretical ceiling established by using ground-truth motion features (`Oracle`), achieving 97.2% of the oracle's Action score.

**Effect of Reference Video Quantity.** Comparing the results for 5 and 9 reference videos in Table 3 shows that more context is beneficial. Increasing the number of references from 5 to 9 boosts the Action score for our method by 1.1 points (`MCT-5` vs. `MCT-9`). In contrast, the improvement for

Table 3: **Comparison of motion adaptation methods.** Our MCT outperforms simpler feature integration strategies and approaches the performance of using ground-truth motion.

| Method | Action↑ | CLIP↑ | DINO↑ | FVD↓ | FID↓ |
|---|---|---|---|---|---|
| Baseline | 53.5 | 91.0 | 85.8 | 88.4 | 10.9 |
| Top-1 | 54.7 | 91.1 | 86.6 | 82.9 | 11.5 |
| Avg-5 | 57.5 | 91.8 | 88.1 | 78.2 | 11.1 |
| Avg-9 | 57.9 | 92.0 | 88.3 | 78.7 | 10.9 |
| MCT-5 | 61.0 | 91.4 | 87.2 | 78.1 | 11.1 |
| **MCT-9 (Ours)** | **62.1** | **92.3** | **88.4** | **69.0** | **9.7** |
| Oracle (GT) | 63.9 | 90.6 | 85.3 | 71.5 | 10.7 |

Table 4: **Robustness and Generalization.** Our method demonstrates strong robustness to noisy (random) retrieval and generalizes effectively to an out-of-distribution (OOD) retrieval database.

| Method | Action↑ | CLIP↑ | DINO↑ | FVD↓ | FID↓ |
|---|---|---|---|---|---|
| Baseline | 53.5 | 91.0 | 85.8 | 88.4 | 10.9 |
| Rand-1 | 52.2 | 90.3 | 85.5 | 88.4 | 11.9 |
| Avg-Rand-9 | 56.4 | **92.2** | **88.7** | 87.7 | 11.0 |
| MCT-Rand-9 | **60.7** | 91.6 | 87.6 | **77.5** | **10.9** |
| MCT-OpenVid-9 | 62.1 | 92.3 | 88.4 | 69.0 | 9.7 |
| MCT-InternVid-9 | 60.9 | 91.7 | 87.4 | 70.7 | 10.5 |

simple averaging is minimal (+0.4 for `Avg-5` vs. `Avg-9`), indicating that our CAMA module is more effective at leveraging richer contextual information to refine motion synthesis.

**Robustness to Noisy Retrieval.** To rigorously test the model's robustness, we simulate a worst-case scenario with noisy retrieval by providing completely random videos as context, as shown in Table 4. While using a single random video (`Rand-1`) slightly degrades performance, our CAMA module (`MCT-Rand-9`) still achieves a remarkable Action score of 60.7, far surpassing the baseline (53.5). This demonstrates that CAMA is not fragile; when faced with irrelevant context, it relies on its strong, learned priors of plausible motion to generate a high-quality result, effectively filtering out noise.

**Generalization to Out-of-Distribution Data.** We evaluate out-of-distribution (OOD) generalization by switching the retrieval database from the in-domain OpenVid-1M [38] to the large-scale InternVid-10M [41] dataset, which has a significantly different data distribution. As seen in Table 4, the performance of `MCT-InternVid-9` remains exceptionally strong (Action 60.9, FVD 70.7), closely tracking the in-distribution results. This demonstrates that our framework is not simply interpolating between similar examples within a familiar dataset. Instead, it has learned to extract and adapt fundamental, transferable motion principles, making it a truly generalizable solution.

### 4.6 Qualitative Results

Figure 4 compares our retrieval-augmented approach against baseline models across diverse scenarios. The results show consistent improvements in motion plausibility and temporal coherence.

When enhancing Dynamicrafter, our method corrects fundamental motion errors. For example, it transforms the unphysical floating of the Newton's cradle balls into a realistic swinging motion. It also animates subjects that are nearly static in the baseline, instilling natural movement in the gesturing man and the jumping person. Similar improvements are seen with CogVideoX. Our approach induces clear progressive motion for the tram, which is largely static and distorted in the baseline video. It also replaces the unnatural "flickering" motion of the runner and the horse with proper gait cycles.

These qualitative results, consistent with our quantitative findings, demonstrate our method's ability to transfer motion semantics to generate more convincing videos. The approach proves especially effective for complex physical phenomena, biological motion, and expressive human actions. Visualizations of the retrieved reference videos that guide the generation process, along with more video results, are available in our supplementary material and on our project website.

Despite its robustness, our method can falter when retrieved videos contain directly conflicting motion cues. For example, to generate a "person jumping," the retrieved set might include both the upward and downward phases of a jump. In such cases, the model may "average" these opposing motions, resulting in a nearly static video with minimal vertical movement. This outcome is a known characteristic of models trained with objectives like MSE, which tend to find a mean solution when faced with conflicting targets.

## 5 Conclusion and Limitations

We have introduced MotionRAG, a novel retrieval-augmented framework that enhances motion realism in image-to-video generation by effectively transferring motion patterns across domains.

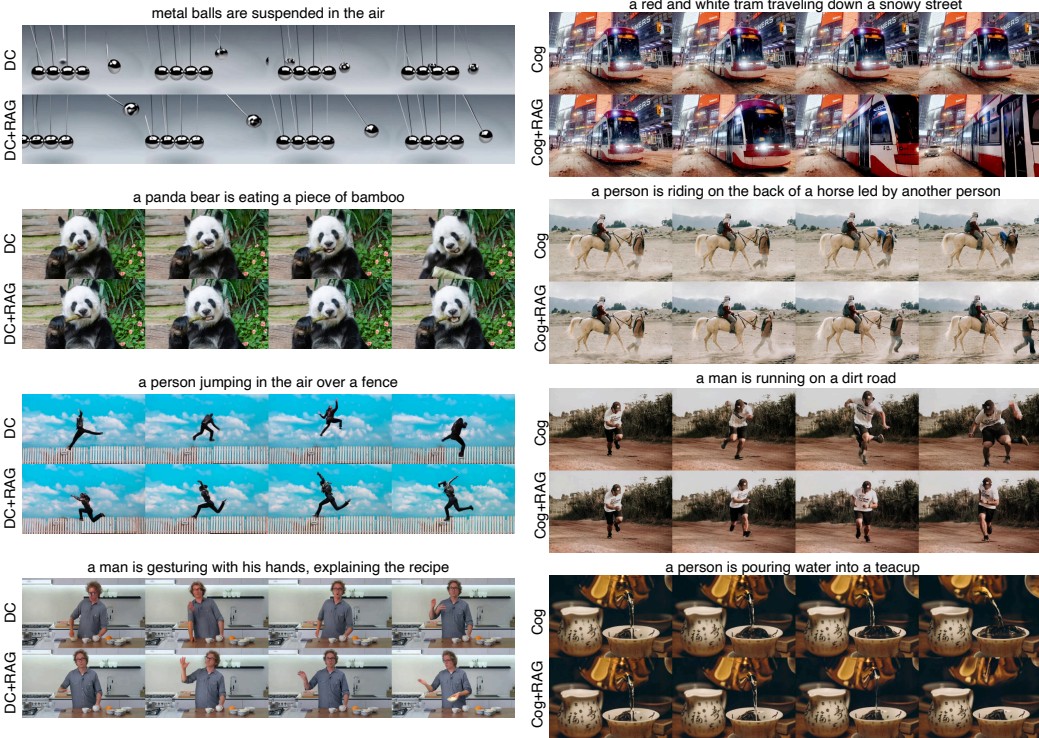

Figure 4: **Qualitative comparison between baseline models and our retrieval-augmented approach across diverse scenarios.** Our method generates more physically plausible and coherent motion, such as realistic object physics, natural animal/human movements, and corrects static or artifacts found in baseline models. Video results are available at our website.

Our Context-Aware Motion Adaptation module formulates motion transfer as an in-context learning problem, enabling robust adaptation without domain-specific fine-tuning. Experiments demonstrate the framework's strong robustness and zero-shot transfer capabilities, showing its consistent improvements across different base models and domains with negligible inference overhead.

Despite these promising results, several limitations still remain. First, a primary limitation is the potential for motion cancellation when retrieved references contain contradictory movements, which can lead to static-like outputs. In addition, while our metrics indicate improved semantic motion, quantitatively evaluating physical plausibility in open-domain videos remains a challenging open research problem. In fact, the development of robust, automated metrics for physical realism is an important issue that would benefit the entire field. On a more practical note, our current framework, constrained by the base models, is designed for short clips, though it could be extended to longer videos by integrating it with models possessing stronger temporal modeling capabilities. Finally, generating fine-grained motions, such as subtle hand gestures, would require both a more specialized retrieval corpus and a video encoder with higher spatiotemporal resolution, marking a clear path for enhancing motion detail in future iterations.

From a broader perspective, our research demonstrates the potential of integrating retrieval mechanisms into generative models, establishing a promising direction for enhancing video generation beyond end-to-end training alone. We believe this retrieval-augmented paradigm represents an important step toward more realistic and controllable video generation systems that can effectively leverage existing motion knowledge.

# Acknowledgement

This work is supported by the National Key R&D Program of China (No. 2022ZD0160900), the Basic Research Program of Jiangsu (No. BK20250009), and the Collaborative Innovation Center of Novel Software Technology and Industrialization.

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

# A Implementation Details

This section provides comprehensive technical details about our MotionRAG framework implementation, covering network architectures, training procedures, and inference pipeline configurations.

**Video and Image Encoders.** We employ VideoMAE-Base [30] pre-trained on Something-Something v2 [46] as our video encoder. We process 16 frames at 224×224 resolution and extract features from all tokens of the final layer. For image encoding, we utilize DINOv2-Large [32], which employs a ViT-L/14 architecture with a hidden dimension of 1024.

**Resamplers.** Our framework employs two separate resamplers for motion and appearance features. These resamplers compress the encoder outputs into a compact set of tokens for efficient processing. The configuration details are provided in Table 5.

Table 5: Configuration for Resamplers.

| Configuration | Motion Resampler | Image Resampler |
|---|---|---|
| Architecture | Transformer | Transformer |
| Layers | 4 | 4 |
| Attention heads | 12 | 12 |
| Hidden dimension | 768 | 768 |
| Feed-forward dimension | 4096 | 4096 |
| Output tokens | 25 | 25 |
| Input feature dimension | 768 (VideoMAE) | 1024 (DINOv2) |
| Output feature dimension | 1024 | 1024 |
| Dropout rate | 0.0 | 0.0 |
| Trainable parameters | 48.0M | 48.3M |
| Initialization | Random | Random |

Table 6: Configuration for Motion Context Transformer.

| Configuration | Motion Context Transformer |
|---|---|
| Architecture | Causal Transformer |
| Layers | 4 |
| Attention heads | 8 |
| Hidden dimension | 1024 |
| Feed-forward dimension | 4096 |
| Maximum sequence length | 500 |
| Attention mask | Block Causal |
| Dropout rate | 0.0 |
| Position embedding | Sinusoid |
| Normalization | LayerNorm |
| Activation | GELU |
| Trainable parameters | 50.4M |

**Motion Context Transformer.** Our Context-Aware Motion Adaptation (CAMA) module uses a causal transformer architecture to facilitate in-context learning for motion transfer. The detailed specifications are provided in Table 6.

**Motion Adapters.** We implement separate Motion Adapters for SVD, DynamiCrafter, and CogVideoX-5b, inserting them after text cross-attention layers in the respective UNet architectures for SVD and DynamiCrafter, and all MMDiT layers for CogVideoX-5b. The configuration details for all adapters are provided in Table 7.

Table 7: Configurations for Motion Adapters across different video generation models.

| Configuration | SVD Motion Adapter | DC Motion Adapter | CogVideoX Motion Adapter |
|---|---|---|---|
| Architecture | Cross-Attention | Cross-Attention | Cross-Attention |
| Insertion points | After text cross-attention layers | After text cross-attention layers | After MMDiT self-attention |
| Number of adapters | 16 | 16 | 42 |
| Attention heads | 8 | 8 | 48 |
| Key/Value dimension | 1024 | 1024 | 3072 |
| Scale factor | 1.0 | 1.0 | 1.0 |
| Trainable parameters | 38M | 38M | 660M |
| Initialization | Random | Random | Random |

**Training Protocol.** We employ a two-stage training approach for our MotionRAG framework across all three models (SVD, Dynamicrafter, and CogVideoX-5b). In the first stage, we train the Motion Adapter and Resampler modules, followed by training the Motion Context Transformer in the second stage. The training hyperparameters for all configurations are detailed in Table 8.

Table 8: Training hyperparameters for the two-stage approach across different models.

| Hyperparameter | Stage 1 (SVD) | Stage 1 (DC) | Stage 1 (Cog) | Stage 2 (Transformer) |
|---|---|---|---|---|
| Dataset | OpenVid-1M [38] | OpenVid-1M [38] | OpenVid-1M [38] | OpenVid-1M [38] |
| Optimizer | AdamW | AdamW | AdamW | AdamW |
| Learning rate | $5 \times 10^{-5}$ | $5 \times 10^{-5}$ | $5 \times 10^{-5}$ | $1 \times 10^{-4}$ |
| Resolution | $320 \times 576$ | $576 \times 1024$ | $480 \times 720$ | $224 \times 224$ |
| Batch size | 16 (2 per GPU) | 16 (2 per GPU) | 8 (1 per GPU) | 64 (8 per GPU) |
| Training steps | 90K | 60K | 60K | 50K |
| Loss function | MSE (denoised prediction) | MSE (denoised prediction) | MSE (denoised prediction) | MSE (motion features) |
| Hardware | 8 NVIDIA RTX A6000 GPUs | 8 NVIDIA RTX A6000 GPUs | 8 NVIDIA RTX A6000 GPUs | 8 NVIDIA RTX A6000 GPUs |
| Training time | 48 hours | 90 hours | 108 hours | 9 hours |

**Video Retrieval System.** Our text-based retrieval system uses GTE-base-1.5-en [39] to encode text queries and video captions into embedding vectors. For all experiments, we retrieve the top-9 most relevant videos based on cosine similarity between text embeddings.

**Video Generation.** We implement our approach on three state-of-the-art image-to-video generation models: Stable-Video-Diffusion-img2vid (SVD) [1], Dynamicrafter-1024 (DC) [2], and CogVideoX-5b-I2V [5]. The hyperparameters used for video generation are provided in Table 9.

Table 9: Generation hyperparameters for OpenVid-1K and SkillVid dataset.

| Hyperparameter | SVD | DC | CogVideoX |
|---|---|---|---|
| Resolution | $576 \times 1024$ | $576 \times 1024$ | $480 \times 720$ |
| Frame count | 16 | 16 | 17 |
| Sampler | EDM | DDIM | DPM |
| Steps | 25 | 30 | 25 |
| CFG scale | 1.0-3.0 | 2.0 | 3 |
| FPS/Motion Strength | 7 | 15 | - |
| Inference time (A6000) | 44s | 90s | 60s |

For video preprocessing during training, we extract 16-frame clips at 8 FPS with random temporal crops during training, and for each video, we use the first frame as the reference image.

# B    Extended Visualization Results

This section presents additional qualitative results generated by our MotionRAG framework across a diverse range of scenarios.

## B.1    Retrieval Visualization

To illustrate how our retrieval mechanism influences motion generation, Figure 5 shows examples of the retrieval process and resulting generated videos. For each query prompt, our system retrieves semantically relevant videos that contain similar motion patterns, which then guide the generation process.

These examples demonstrate how our approach transfers motion characteristics across visual domains:

**Physics-based motion:** For "metal balls suspended in the air" the system retrieves videos of Newton's cradles, magnetic balls, and physics experiments. The generated video exhibits realistic pendulum-like oscillations derived from these references.

**Fluid dynamics:** For "a person pouring water into a teacup" retrieved videos show various pouring actions with different teapots and cups. The generated video captures the natural flow of liquid and the subtle hand movements during pouring.

**Human locomotion:** For "a man running on a dirt road" the system retrieves videos of people jogging in various environments. The generated video reproduces natural running gait and body mechanics.

**Animal-human interaction:** For "a person riding on the back of a horse led by another person" retrieved videos show various horse-riding scenarios. The generated video captures the coordinated movement between horse and riders.

Despite differences in background, lighting, and specific object arrangements, the retrieved videos provide valuable motion priors that guide the generation process. The resulting videos exhibit realistic motion while maintaining the visual appearance specified in the input images.

## B.2    Additional Generation Results

Figure 6 showcases video sequences generated using our Dynamicrafter+RAG and CogVideoX+RAG models, demonstrating their ability to produce realistic motion patterns across various domains.

These results highlight our methods' ability to transfer motion patterns across visual domains while maintaining physical plausibility and semantic consistency. The generated videos preserve the appearance specifications while introducing temporally coherent motion that aligns with the described actions. For the best view, please refer to the videos in our website (https://github.com/MCG-NJU/MotionRAG).

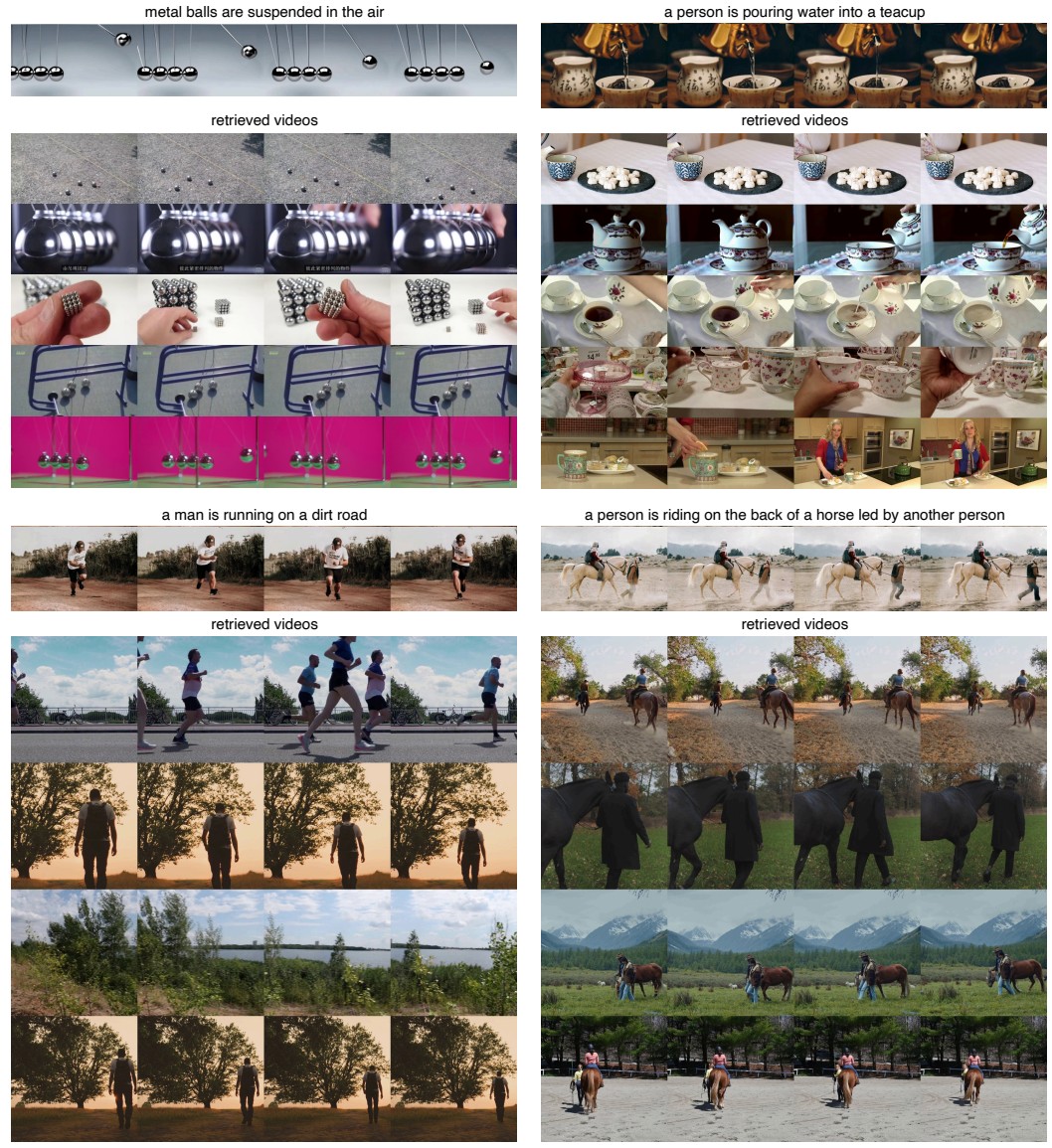

Figure 5: **Retrieval and generation examples.** Each panel shows a different scenario: (top-left) metal balls suspended in air with pendulum-like motion, (top-right) a person pouring water into a teacup, (bottom-left) a man running on a dirt road, and (bottom-right) a person riding on a horse led by another person. For each example, the top row displays frames from our generated video, while the rows below show frames from retrieved reference videos. Note how our system extracts relevant motion patterns from visually different but semantically similar videos.

a motorcycle driving down a road

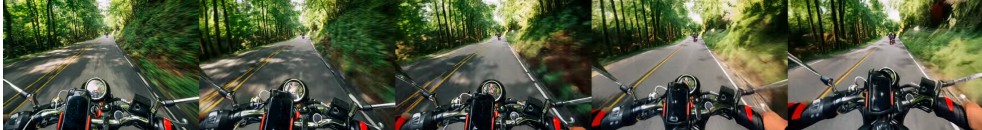

a bunch of cars are driving on a highway

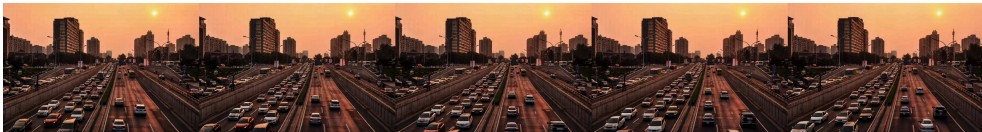

a penguin walking on a beach near the water

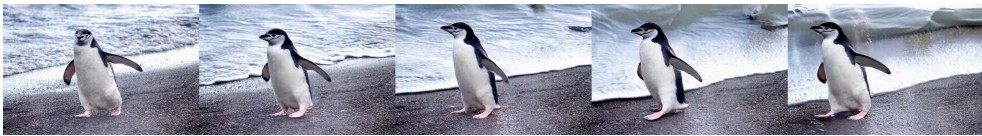

a red double decker bus driving down a street

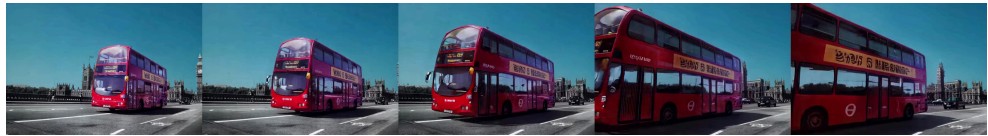

a city bus driving down a snowy street at night

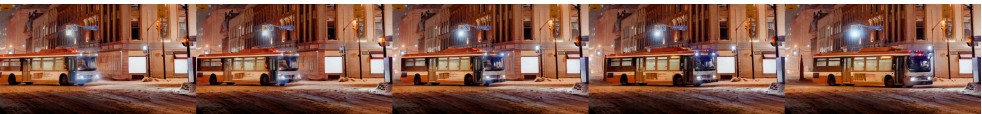

an older man jogging by the water

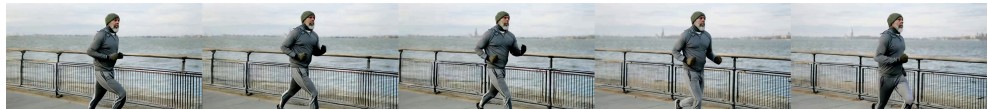

a red panda eating bamboo in a zoo

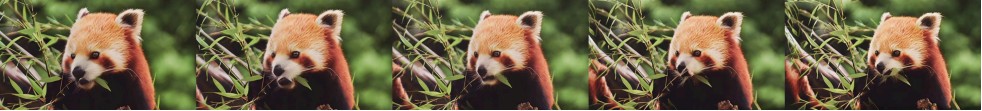

A yellow boat is cruising in front of a bridge

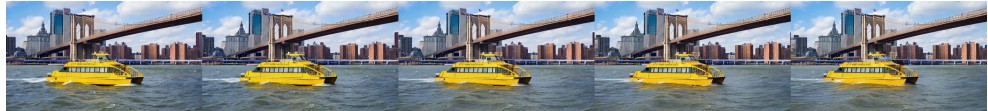

a zebra walking across a dirt road near a field

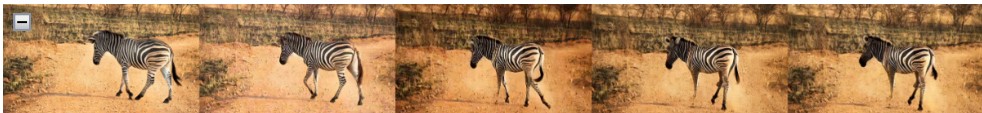

Figure 6: **Additional video generation results.** Each row displays five frames from a generated video sequence. The first four rows show results from CogVideoX+RAG, while the remaining rows present Dynamicrafter+RAG outputs. Our approach successfully captures motion characteristics across these diverse scenarios.

