# OpenReview forum: "MotionRAG: Motion Retrieval-Augmented Image-to-Video Generation"
_NeurIPS.cc/2025/Conference — NeurIPS 2025 poster_

### Official Review · Reviewer_GD23 · 2025-06-29

**Clarity:** 3
**Significance:** 2
**Originality:** 2
**Rating:** 4
**Confidence:** 4

**Summary:**

This paper introduces MotionRAG, a retrieval-augmented framework designed to improve motion realism in image-to-video generation. The authors propose several contributions, including Retrieval-based motion adaptation and Seamless motion injection. The method enhances motion realism across diverse domains without retraining, supporting zero-shot generalization by updating the retrieval database.

**Questions:**

1. Please address the two concerns in the weakness section.
2. Further disussion between the proposed implicit motion conditioned mechanism and explicit optical-flow-based motion condition mechanism (Motion-I2V and MOFA-Video) would be helpful to understand the contribution of this submission.

**Ethical Concerns:**

["NO or VERY MINOR ethics concerns only"]

**Final Justification:**

After rebuttal, I lean towards borderline accept for this submission.

**Limitations:**

yes

**Quality:**

2

**Strengths And Weaknesses:**

Strengths:
- The concept that "motion can be inherently transferred across domains" is well-justified and intuitive.
- The authors test their method on three different base models, demonstrating broad applicability.

Weaknesses:
- The retrieval database and test sets both come from OpenVid-1M and SkillVid, meaning the test videos likely share a similar distribution with the retrieved references. This could inflate performance compared to real-world scenarios where videos may differ significantly from the database. A more convincing evaluation would involve out-of-distribution or in-the-wild videos.
- Lack of physics-based evaluation: While the abstract and introduction emphasizes physical realism, the paper does not quantitatively assess whether generated videos adhere to physical laws. Recent benchmarks (e.g., "Do generative video models understand physical principles? arxiv '25") could provide more rigorous metrics for evaluating motion plausibility and dynamics.

---

> ### Author Rebuttal · Authors · 2025-07-29
>
> We sincerely thank Reviewer GD23 for the constructive feedback. We are glad the reviewer found our core concept intuitive and our multi-model validation a key strength. We would like to address the important concerns raised regarding evaluation and comparison to other methods.
>
> **1. Regarding In-Distribution Evaluation and Out-of-Distribution Testing (Weakness 1).**
>
> This is a very valid point. To rigorously test our method's generalization ability beyond just swapping specialized datasets, we conducted a new **out-of-distribution (OOD) experiment**.
>
> - **New OOD Experiment:** We tested our method on the OpenVid-1K test set while using a completely different, large-scale video understanding dataset, **InternVid-10M**, as our retrieval database. It is crucial to note that InternVid-10M is curated for video *understanding* tasks and has a significantly different data distribution (e.g., content, caption, camera work) compared to video *generation* datasets like OpenVid. The results are as follows (on Dynamicrafter):
>
>   | Retrieval DB | Action↑ | FVD↓ | FID↓ | CLIP↑ | DINO↑ |
>   | --- | --- | --- | --- | --- | --- |
>   | Baseline (None) | 53.5 | 88.4 | 10.9 | 91.0 | 85.8 |
>   | OpenVid-1M (In-Dist.) | **62.1** | **69.0** | **9.7** | **92.3** | **88.4** |
>   | InternVid-10M (OOD) | 60.9 | 70.7 | 10.5 | 91.7 | 87.4 |
>
>
> - **Conclusion:** The results clearly show that even when retrieving from a large-scale, out-of-distribution database, our method still achieves a dramatic improvement over the baseline, with performance nearly on par with using an in-distribution database. This provides strong evidence that our method is not simply "memorizing" patterns from a specific dataset distribution but is capable of robustly generalizing across diverse data sources. We will add this new experiment and analysis to our paper.
>
> **2. Regarding Physics-Based Evaluation.**
>
> We agree that quantitative, physics-based evaluation is an important and emerging area.
>
> - **Alignment with Evaluation Principles:** Our evaluation philosophy is conceptually aligned with the one proposed in the benchmark cited by the reviewer ("Do generative video models understand physical principles?"): both methodologies compare the generated video to a ground truth (GT) video, operating under the assumption that the GT video is physically plausible.
> - **Suitability of Metrics for Open-Domain Content:** The primary difference lies in the specific metrics used, which reflects the different nature of the evaluation data. The cited work employs low-level metrics like Spatial IoU and MSE, which are highly effective for their dataset of **simple, single-object interactions in controlled settings** (e.g., a ball bouncing). However, these coordinate-based metrics are less suitable for the **complex, open-domain scenarios** in our evaluation, which often feature articulated human/animal motion, multiple interacting objects, and unconstrained camera movement. In such cases, these low-level metrics could penalize a generated video for being physically plausible but not pixel-perfectly identical to the GT (e.g., a slightly different but equally valid running gait).
> - **Our Metrics Capture Semantic Plausibility:** For this reason, we chose a suite of metrics better suited for open-domain content. The Action Score, in particular, uses a high-level action recognition model to assess if the *semantic nature* of the motion is correct (e.g., does it look like "running" or "jumping"?). The consistent and significant improvements across our entire suite provide strong, holistic evidence of enhanced motion realism.
>
> **3. Regarding Comparison with Explicit Motion-Conditioned Methods.**
>
> This is an excellent question that helps clarify our contribution.
>
> - **High-Level Semantics vs. Low-Level Pixels:** The key difference is the level of abstraction. Methods like Motion-I2V rely on **low-level, pixel-based motion representations like optical flow**. While precise, optical flow is **tightly coupled with appearance and structure**, making it very difficult to transfer across domains (e.g., the optical flow of a person walking cannot be easily applied to a robot walking). It is also **computationally expensive** to generate.
> - **Our Advantage: Cross-Domain Transfer and Efficiency.** Our approach uses **high-level, semantic motion features** that capture the *essence* of a motion (e.g., the concept of "walking"). This abstraction is what enables **robust cross-domain transfer** and is the core reason our method works. Furthermore, our motion feature extraction is **highly efficient**, taking less than one second. We will expand the related work section to explicitly detail these key differences.
>
> We believe these clarifications, supported by new experimental evidence, thoroughly address the reviewer's concerns. We appreciate the feedback, which has prompted us to further strengthen our paper's evaluation and positioning.

---

> > ### Author Response · Authors · 2025-08-05
> >
> > Dear Reviewer(GD23): We would deeply value the opportunity to continue our discussion with you. Your insights and additional questions would be invaluable in further refining the quality of MotionRAG, and we sincerely look forward to your ongoing feedback.

---

> ### Comment · Reviewer_GD23 · 2025-08-05
> **Reply to the authors' rebuttal**
>
> 1. In-Distribution Evaluation and Out-of-Distribution Testing
>
> I find the authors’ response to this concern to be convincing.
>
> 2. Comparison with Explicit Motion-Conditioned Methods
>
> The additional discussion sufficiently addresses my concerns. I believe this discussion could be incorporated into the paper to better emphasize the contributions of the work.
>
> 3. Physics-Based Evaluation
>
> While I understand the authors’ rationale, I remain somewhat concerned about the lack of explicit evaluation on the physics-based aspects.
>
> Given these considerations, I lean towards borderline accept for this submission.

---

> > ### Author Response · Authors · 2025-08-05
> >
> > Dear Reviewer GD23,
> >
> > Thank you for your time and for the thoughtful re-evaluation of our work. We are pleased that our out-of-distribution experiments and the clarified comparison to explicit motion-conditioned methods have addressed your concerns.
> >
> > We also appreciate your persistent and valuable point regarding physics-based evaluation. We agree that this is a crucial and challenging direction for the entire field, and we consider it an important avenue for future research.
> >
> > As promised, we will integrate the detailed discussion comparing our approach to explicit motion-conditioned methods into the final manuscript to better highlight our contributions.
> >
> > Thank you once again for your constructive feedback and your support for our submission.

---

### Official Review · Reviewer_rjfD · 2025-07-02

**Clarity:** 3
**Significance:** 3
**Originality:** 3
**Rating:** 4
**Confidence:** 4

**Summary:**

This paper proposes MotionRAG, a retrieval-augmented framework for video generation that enhances motion realism by transferring motion priors from semantically similar videos retrieved via text prompts.
Experiments across general and domain-specific datasets (OpenVid-1K and SkillVid) show MotionRAG consistently improves motion realism, semantic alignment, and video quality with minimal computation overhead. Ablations further validate the design.

**Questions:**

See Weaknesses.

**Ethical Concerns:**

["NO or VERY MINOR ethics concerns only"]

**Limitations:**

Yes

**Paper Formatting Concerns:**

The authors use blue-colored fonts in the title, which might implicitly suggest an affiliation with organizations such as Meta, Seed Group in Bytedance, or Ant Group. Of course, this does not affect my judgment, so I do not believe it violates the double-blind review policy. However, I would encourage the authors to avoid using such suggestive color schemes in the submission stage.

**Quality:**

3

**Strengths And Weaknesses:**

stengths:
1. This paper is well-written and easy to understand.
2. The paer propose a RAG based video generation method which is interesting and meaningful for video generation community.
3. The expertiments are extensive enough to demonstrate the performance.

weaknesses:
One of the potential weaknesses of the proposed method is its strong dependence on the quality of the retrieved reference videos. It is unclear whether the model will be able to automatically filter or reduce the impact of these sub-optimal references if there are errors in the search results. Although the Motion Context Transformer is designed to have context-aware motion adaptation capabilities, its robustness to low-quality or erroneous search results is not explicitly analyzed in this paper, and mechanisms such as confidence weighting or retrieval screening are not proposed to alleviate this problem. Further exploration of denoising or robustness mechanisms related to retrieval quality will help improve the reliability of the system in practical applications.

---

> ### Author Rebuttal · Authors · 2025-07-29
>
> We sincerely thank Reviewer rjfD for the positive and encouraging review. We are delighted that the reviewer found our paper well-written, our approach interesting and meaningful, and our experiments extensive. The reviewer raised a very important point about the system's robustness to sub-optimal retrieval results, which we would like to address comprehensively.
>
> **Regarding the Robustness to Low-Quality or Erroneous Search Results.**
>
> We agree that robustness to imperfect retrieval is critical for any RAG-based system's practical application. While we did not explicitly frame it as a "denoising" mechanism, our Context-Aware Motion Adaptation (CAMA) module is inherently designed to be robust to such scenarios, thanks to the motion priors it learns during training. Our ablation studies provide strong, quantitative evidence for this capability.
>
> 1. **CAMA's Inherent Filtering Capability through Learned Priors:** The reviewer correctly identifies that we did not add an explicit filtering or weighting mechanism. This is by design. Our CAMA module, a causal transformer, learns to perform this function implicitly. By training on millions of diverse video-text pairs, CAMA develops a strong internal prior of plausible physical motion. When processing a sequence of multiple retrieved examples, it leverages this prior to identify and amplify common, relevant motion patterns while down-weighting or ignoring outlier features from sub-optimal references. It learns to reason about motion, not just copy it.
>
> 2. **Strongest Evidence: Robustness to Random Retrieval (Worst-Case Noise).** The most compelling evidence of this robustness is presented in our ablation study in Table 4. The "MCT-Rand-9" experiment uses 9 completely random videos as context—the worst-case scenario for retrieval quality. Even with this entirely irrelevant context, our method's performance is remarkable.
>
>   | Method | Action↑ | FVD↓ | FID↓ | CLIP↑ | DINO↑ |
>   | --- | --- | --- | --- | --- | --- |
>   | Baseline | 53.5 | 88.4 | 10.9 | 91.0 | 85.8 |
>   | MCT-9 (Ours) | **62.1** | **69.0** | **9.7** | **92.3** | **88.4** |
>   | MCT-Rand-9 | 60.7 | 77.5 | 10.9 | 91.6 | 87.6 |
>
>   As the table shows, the Action score for "MCT-Rand-9" (60.7) is not only far superior to the baseline (53.5) but is also impressively close to the performance with high-quality retrieval ("MCT-9", 62.1). This directly demonstrates that CAMA is not fragile. When faced with erroneous inputs, it falls back on its powerful learned priors about how things should move, ensuring a plausible and high-quality output. This directly addresses the reviewer's core concern.
>
>
> Regarding the **Paper Formatting Concerns**, we thank the reviewer for pointing this out. The color choice was purely for aesthetic reasons and had no intention of implying any affiliation.
>
> We are confident that these clarifications, supported by direct quantitative evidence from our experiments, fully address the reviewer's valid concerns and further strengthen the paper. We thank the reviewer again for their positive assessment and constructive feedback.

---

> > ### Comment · Reviewer_rjfD · 2025-08-04
> >
> > Most of my concerns have been addressed, and I will keep my rating

---

> > > ### Author Response · Authors · 2025-08-07
> > >
> > > Dear Reviewer rjfD,
> > >
> > > Thank you for your time and for considering our rebuttal. We are glad to hear that our clarifications have addressed most of your concerns.
> > >
> > > We appreciate your valuable feedback throughout this process.
> > >
> > > Thank you again.

---

### Official Review · Reviewer_SsZX · 2025-07-03

**Clarity:** 3
**Significance:** 2
**Originality:** 3
**Rating:** 4
**Confidence:** 4

**Summary:**

This manuscript presents MotionRAG, a retrieval-augmented framework designed to improve motion realism in image-to-video generation. The core contributions are:

-A pipeline that uses sentence embeddings to retrieve semantically relevant reference videos from a large corpus, providing real-world motion priors for generation.
-Lightweight adapters inserted after cross-attention layers in frozen diffusion models, seamlessly integrating adapted motion features to guide video synthesis.

**Questions:**

see comments above.
It will be necessary to address the comments above to maintain positive score.

**Ethical Concerns:**

["NO or VERY MINOR ethics concerns only"]

**Final Justification:**

After going through the response, the reviewer tends to maintain the original score as it is.

**Limitations:**

see comments above

**Paper Formatting Concerns:**

see comments above

**Quality:**

3

**Strengths And Weaknesses:**

Strengths
-By explicitly leveraging retrieved video examples, MotionRAG grounds motion dynamics in real data
-The framework’s modular design allows for zero-shot transfer.
-The paper thoroughly investigates retrieval quality (random vs. semantic), number of references (5 vs. 9), and integration strategies (Top-1, Avg-9, MCT-9)

Weaknesses:
-Experiments focus on short clips (~2 seconds) and a fixed set of general and instructional domains; the approach’s scalability to long-range motions or fine-grained action nuances (e.g., subtle gestures) remains untested.
-Large-scale retrieval (e.g., nearest-neighbor search over millions of embeddings) can be nontrivial; reporting end-to-end latency including retrieval indexing would be necessary.
-MotionRAG’s performance hinges on the diversity and relevance of the retrieval corpus; can rare or unique motions perform well? better to add analysis and failure cases
-Will the work be open-source or public? It will be necessary to provide temporary/anonymous repo and demo code during the review.

---

> ### Author Rebuttal · Authors · 2025-07-29
>
> We sincerely thank Reviewer SsZX for the positive and insightful review. We are delighted that the reviewer recognized the effectiveness of our retrieval-based approach, the flexibility of our modular design, and the thoroughness of our experimental investigation. We appreciate the constructive feedback and would like to address the raised points to further strengthen our paper.
>
> **1. Regarding Scalability to Longer Videos and Fine-Grained Motions (Weakness 1).**
>
> We agree that extending our work is a valuable future direction.
>
> - **Current Scope on Short Clips:** Our experiments focus on ~2 second clips, which is inherent architectural limitations of the baseline models we used, such as SVD and DynamiCrafter, which are themselves designed primarily for short video generation. Within this established setting, we have demonstrated significant improvements.
> - **Potential for Longer Videos:** Our framework is not fundamentally limited to short clips. When integrated with base models possessing stronger temporal modeling capabilities, such as CogVideoX, we observed that our approach can readily extend to generate coherent videos of up to 6 seconds. We focused on 2-second clips in our reported results primarily due to computational constraints but will mention this extensibility in the paper.
> - **Fine-Grained Motions:** We agree that fine-grained motions (e.g., subtle gestures) pose a challenge. Performance would depend on both a specialized retrieval corpus containing such examples and a video encoder capable of capturing these nuances. We will explicitly state these points as promising avenues for future research in our Limitations section.
>
> **2. Regarding End-to-End Latency Including Retrieval (Weakness 2).**
>
> This is an excellent point regarding practical deployment. Our framework is designed for high efficiency.
>
> - **Highly Efficient Retrieval:** We use LanceDB, a modern, high-performance vector database. On our CPU, retrieving the top-9 examples from our 1 million entry database takes only 40ms on average. This is highly scalable; even if the database were expanded to 10 million entries, the retrieval time would only increase to approximately 100ms. This confirms that retrieval is not a bottleneck.
>
> **3. Regarding Dependency on the Retrieval Corpus and Failure Cases (Weakness 3).**
>
> We agree that analyzing the performance boundaries is crucial.
>
> - **Generalization to Rare Motions:** The performance on rare motions is not solely dependent on finding a perfect match in the database. During its training on a large and diverse dataset, our CAMA module learns a generalized prior of physical motion and object-scene interactions. This means that for a rare motion, even if the retrieved videos are only loosely related, CAMA can leverage this learned prior to synthesize a plausible motion.
>
> - **Robustness to Imperfect Retrieval:** As shown in our ablation study (Table 4, "MCT-Rand-9"), our CAMA module can distill meaningful motion priors even from randomly selected videos, demonstrating a degree of robustness.
>
> - **Failure Case Analysis:** The primary failure mode we identified occurs when retrieved videos contain directly opposing motions. For example, when generating a video of "a person jumping up," the retrieval set may contain both the upward and downward phases of a jump. In this case, our CAMA module might average these conflicting motion vectors, resulting in a nearly static output where the person barely moves. This highlights a limitation in resolving contradictory motion priors.
>
>
> **4. Regarding Open-Sourcing (Weakness 4).**
>
> - **Commitment to Open Source:** We are fully committed to open science. We will make our code and pre-trained model weights publicly available upon publication to ensure full reproducibility.
>
> We believe that by incorporating these clarifications, additional analyses, and a commitment to releasing our code, we have thoroughly addressed the reviewer's constructive points. We thank the reviewer again for their valuable feedback, which will significantly improve our paper.

---

### Official Review · Reviewer_v6PA · 2025-07-03

**Clarity:** 3
**Significance:** 2
**Originality:** 2
**Rating:** 4
**Confidence:** 4

**Summary:**

This paper proposes a retrieval-augmented framework, MotionRAG, to enhance motion realism in video generation by transferring motion priors from relevant reference videos. The framework addresses the challenge of generating physically plausible and semantically coherent motion. MotionRAG contains a key component, Context-Aware Motion Adaptation (CAMA), that adapts retrieved motion priors to the target image context with in-context learning. Experiments demonstrate that MotionRAG can improve motion quality across multiple base models and domains with negligible computational overhead.

**Questions:**

- The captions given by the existing video captioners are bound to be rough and unable to accurately describe the physical motions. When constructing a dataset, is it necessary to manually select homogeneous reference videos?
- I highly doubt the robustness of filtering out reference videos based solely on text similarity. Text cannot convey the details of physical motion and may lead to incorrect retrieval. Even for the examples shown in the supplementary material, the text descriptions are inaccurate. There are also large differences in physical motion between the retrieved reference videos. What about the reference video retrieved when the text description is out of domain?

**Ethical Concerns:**

["NO or VERY MINOR ethics concerns only"]

**Final Justification:**

Most of my concerns have been addressed in the rebuttal.
- The dataset construction is automated and scalable.
- MotionRAG outperforms the baseline when processing OOD data.

But the experiments related to physical constraints are still insufficient.

**Limitations:**

I would suggest adding more analysis and visualization of the dataset in the revised version.

And more metrics for measuring physical motion need to be considered.

**Quality:**

2

**Strengths And Weaknesses:**

Strengths
- This paper applies the RAG technique to video generation to improve motion realism, which is meaningful for the video generation task.
- This paper achieves zero-shot effective motion transfer across different domains with negligible computational overhead.
- The writing of this paper is clear and easy for the reader to follow.

Weaknesses
- The RAG technique itself is not novel and has been proven to be effective in other related generation tasks. Therefore, the contribution of applying RAG to motion transfer is relatively weak.
- One of the crucial points of RAG is to construct the dataset, but this paper lacks the necessary data analysis. How such datasets are constructed is important to the community.
- This paper lacks a detailed analysis and visualization of the semantic similarity of the retrieved reference videos. Based on the current version, MotionRAG relies on the existence of videos with highly similar motion in the dataset and cannot accurately achieve motion transfer.
- Existing evaluation metrics cannot fully reflect the quality of motion, especially the physical constraints claimed in this paper.

---

> ### Author Rebuttal · Authors · 2025-07-30
>
> We sincerely thank Reviewer v6PA for the detailed and constructive feedback. We appreciate that the reviewer recognized the significance of our work in improving motion realism, its zero-shot capabilities, and the clarity of our writing. We would like to address the raised concerns, which we believe stem from a potential misunderstanding of our core technical contribution—particularly the robustness and learned priors of our CAMA module.
>
> **1. Regarding Novelty: The Core Innovation is CAMA's In-Context Adaptation, Not Just the Application of RAG.**
>
> We agree that RAG is an established paradigm. However, our main contribution is not simply *applying* RAG, but proposing a novel method for how motion priors are adapted and synthesized from imperfect, real-world data. This is the specific function of our Context-Aware Motion Adaptation (CAMA) module.
>
> Our ablation study in Table 3 provides direct evidence of this. A naive RAG approach that simply retrieves and averages features ("Avg-9") offers only a moderate improvement, yielding an Action score of 57.9 and an FVD of 78.7. In contrast, our CAMA module ("MCT-9") achieves significantly superior results, with an Action score of 62.1 and a much lower FVD of 69.0. It demonstrates that the innovation lies in the intelligent, in-context synthesis of motion, not just the retrieval framework. CAMA learns to *adapt* motion by reasoning over multiple examples, a far more complex and novel task than simple feature blending.
>
> **2. Regarding Dataset Construction and Robustness to Imperfect Retrieval.**
>
> This is the reviewer's primary concern, and we thank them for the insightful questions. Our framework is explicitly designed to be robust to imperfect and diverse retrieval results, which is a key strength that we wish to clarify.
>
> - **Dataset Construction is Automated and Scalable:** We apologize for not making this clearer. A key strength of our method is that it does not require any manual curation or selection of reference videos. For the OpenVid-1M dataset, we used an Llama3-8B model in a one-time, automated preprocessing step to generate concise motion-focused descriptions from the original detailed captions. This makes our approach practical, easily reproducible, and scalable.
>
> - **CAMA's Robustness is Due to Learned Priors:** The reviewer rightly doubts the reliability of text-based retrieval. Our CAMA module is specifically designed to handle this challenge. During its training on millions of video, CAMA learns a powerful internal prior about plausible motion for given objects and scenes. This learned knowledge makes it robust, as demonstrated by the following experiments:
>
>   | Retrieval Method | Action↑ | FVD↓ | FID↓ | CLIP↑ | DINO↑ |
>   | --- | --- | --- | --- | --- | --- |
>   | No Retrieval (Baseline) | 53.5 | 88.4 | 10.9 | 91.0 | 85.8 |
>   | Random Retrieval | 60.7 | 77.5 | 10.9 | 91.6 | 87.6 |
>   | OOD Retrieval | 60.9 | 70.7 | 10.5 | 91.7 | 87.4 |
>   | In-Distribution Retrieval | **62.1** | **69.0** | **9.7** | **92.3** | **88.4** |
>
>   The data above reveals two critical insights into CAMA's robustness:
>
>   1. **It excels even with random, irrelevant context.** The "Random Retrieval" row represents the worst-case scenario. Even here, CAMA dramatically improves the Action score from 53.5 to 60.7 and reduces the FVD from 88.4 to 77.5. This proves that CAMA does not blindly trust its inputs. Instead, it falls back on its strong, learned priors to generate a physically plausible motion.
>   2. **It generalizes across different data distributions.** The "OOD Retrieval" experiment serves as a stringent test of this capability. We retrieved from InternVid-10M, a large-scale dataset curated for video *understanding*, whose distribution (content, style, captions) differs substantially from our generation-focused dataset. Even with this domain shift, performance remains remarkably strong (Action 60.9, FVD 70.7), closely tracking the in-distribution results. It proves that CAMA is not merely interpolating between similar examples but has learned to extract and adapt fundamental motion principles, even from a out-of-distribution source. This directly addresses the concern that our method might be fragile and overly reliant on the retrieval database's domain.
>
> In essence, CAMA does not simply copy motion; it *infers* it, using retrieved examples as context but relying on its own learned understanding of motion physics and semantics to produce a robust final result.
>
> **3. Regarding Evaluation Metrics and Physical Plausibility.**
>
> We concur with the reviewer that quantitatively measuring physical plausibility is a challenging, open research problem.
>
> Our evaluation philosophy is conceptually aligned with recent, dedicated work on this topic (e.g., Motamed et al., "Do generative video models understand physical principles?", 2025). Both our approach and theirs rely on the core principle of comparing a generated video to a ground-truth (GT) video, under the assumption that the GT video is physically plausible.
>
> The primary difference lies in the specific metrics chosen, which reflects the different nature of the evaluation data.
>
> - Metrics like Spatial/Spatiotemporal IoU and MSE, as used in the cited work, are highly effective for their dataset of simple, single-object interactions in controlled settings.
> - However, these low-level, coordinate-based metrics are not robust for the complex, open-domain scenarios in our evaluation, which often feature articulated human/animal motion, multiple interacting objects, and camera movement. In these cases, a minor, physically plausible variation (like a slightly different camera angle or running gait) could unfairly penalize a low-level metric score.
> - Therefore, we opted for a suite of metrics that are better suited for our setting. The Action Score, in particular, uses a high-level action recognition model to assess if the *semantic* nature of the motion is correct. The consistent and significant improvements across our entire suite of metrics (Action Score, FVD, etc.) provide strong, holistic evidence of enhanced motion quality.
>
> In summary, we believe the reviewer's primary concerns are centered on the perceived fragility of a retrieval system. We hope our explanation clarifies that our core contribution, CAMA, is a robust motion adaptation mechanism that leverages in-context learning to thrive on diverse, imperfect, and even out-of-distribution reference data. We respectfully request the reviewer to reconsider their evaluation in light of these clarifications.

---

> > ### Author Response · Authors · 2025-08-05
> >
> > Dear Reviewer(v6PA): We would deeply value the opportunity to continue our discussion with you. Your insights and additional questions would be invaluable in further refining the quality of MotionRAG, and we sincerely look forward to your ongoing feedback.

---

> ### Comment · Area_Chair_DK65 · 2025-08-05
> **Feedback Needed - Your AC**
>
> Dear Reviewer v6PA,
>
> I notice that the authors have submitted a rebuttal. Could you please let me know if the rebuttal addresses your concerns? Your engagement is crucial to this work.
>
> Thanks for your contribution to our community.
>
> Your AC

---

> ### Comment · Reviewer_v6PA · 2025-08-06
>
> Thanks for the insightful responses.
>
> 1. Regarding Novelty
>
> I admit the role of CAMA's In-Context Adaptation. I just want to point out that the first contribution claimed by this paper is insufficient but acceptable.
>
> 2. Regarding Dataset Construction and Robustness to Imperfect Retrieval
>
> The construction of the dataset is now clear. And the experiment results under extreme OOD conditions are convincing.
>
> 3. Regarding Evaluation Metrics and Physical Plausibility
>
> I understand that evaluating physical constraints is challenging. But the metrics, like FID and FVD, are relatively simple and only provide a rough evaluation of the quality of motion generation. I still recommend adding experiments or demos on physical constraints in the final version.
>
> I think most of my concerns have been addressed. I decided to change my score to borderline accept.

---

> > ### Author Response · Authors · 2025-08-07
> >
> > Dear Reviewer v6PA,
> >
> > Thank you for your detailed feedback and for taking the time to review our response. We are very pleased to hear that our clarifications regarding the novelty of CAMA and the robustness of our system were convincing.
> >
> > We also sincerely appreciate your valuable perspective on the evaluation of physical plausibility. We agree that this is a critical area and your recommendation to include further analysis is well-taken. We will certainly consider how to best incorporate additional demonstrations or discussions on this topic in our final version.
> >
> > Thank you once again for your constructive engagement and for raising your score.

---

### Note · Authors · 2025-08-12

We are sincerely grateful to the reviewers and the Area Chair for their insightful and constructive feedback. The discussions have been invaluable in strengthening our paper.

The primary concern centered on our framework's robustness to imperfect retrieval. We addressed this by highlighting evidence from our original ablation study on random retrieval (a worst-case scenario) and supplementing it with a new, stringent out-of-distribution (OOD) test using the InternVid-10M dataset. These combined results demonstrated that our core module, CAMA, maintains strong performance even with irrelevant context, which successfully convinced the reviewers.

We also clarified our core contributions. We emphasized that our novelty lies not in simply applying RAG, but in our Context-Aware Motion Adaptation (CAMA) module. Furthermore, we distinguished our approach from methods using explicit motion conditioning by explaining that our use of high-level semantic motion features enables robust cross-domain transfer and is more efficient than low-level, pixel-based representations like optical flow. Regarding evaluation, we acknowledged the challenges in measuring physical plausibility and contextualized our choice of metrics as being more suitable for the complex, open-domain scenarios we address.

We are encouraged that our responses and new evidence were well-received, leading Reviewers v6PA and GD23 to raise their recommendations. In the final version, we will incorporate these key improvements, as promised in our individual responses. To facilitate reproducibility and future research, we will make our code and models publicly available.

Thank you again for your consideration.

---

### Decision · Program_Chairs · 2025-09-17

**Decision:**

Accept (poster)

**Comment:**

All the reviewers unanimously voted for acceptance. After checking the review, the rebuttal, the manuscript, and the discussion, the AC recommended acceptance on the condition that the manuscript would be revised in the camera-ready version by taking the reviewers' suggestions into consideration, such as the experiments on physical constraints, code open source, etc.